# Essential Nanostructure Parameters to Govern Reinforcement and Functionality of Poly(lactic) Acid Nanocomposites with Graphene and Carbon Nanotubes for 3D Printing Application

**DOI:** 10.3390/polym12061208

**Published:** 2020-05-26

**Authors:** Rumiana Kotsilkova, Evgeni Ivanov, Vladimir Georgiev, Radost Ivanova, Dzhihan Menseidov, Todor Batakliev, Verislav Angelov, Hesheng Xia, Yinghong Chen, Dzmitry Bychanok, Polina Kuzhir, Rosa Di Maio, Clara Silvestre, Sossio Cimmino

**Affiliations:** 1Institute of Mechanics (OLEM), Bulgarian Academy of Sciences, Sofia 1113, Bulgaria; ivanov_evgeni@yahoo.com (E.I.); shtastie.ivanova@gmail.com (R.I.); cihan.menseidov@gmail.com (D.M.); todorbat@gmail.com (T.B.); verislav@abv.bg (V.A.); 2NanoTechLab Ltd., Sofia 1113, Bulgaria; vladofe@gmail.com; 3State Key Laboratory of Polymer Materials Engineering, Polymer Research Institute of Sichuan University, Chengdu 610065, China; xiahs@scu.edu.cn (H.X.); johnchen@scu.edu.cn (Y.C.); 4Institute of Nuclear Physics, Belarus State University, 220006 Minsk, Belarus; dzmitrybychanok@yandex.by (D.B.); polina.kuzhir@uef.fi (P.K.); 5Tomsk State University, Tomsk 634050, Russia; 6Institute of Photonics, University of Eastern Finland, FI-80101 Joensuu, Finland; 7Institute of Polymers, Composites and Biopolymers, CNR, 80078 Pozzuoli (NA), Italy; rosadimaio1988@libero.it (R.D.M.); clara.silvestre@ipcb.cnr.it (C.S.); sossio.cimmino@ipcb.cnr.it (S.C.)

**Keywords:** degree of dispersion, percolation threshold, interfacial interactions, homogeneous network, segregated network, aggregated structure, tensile properties, electrical and thermal conductivity, electromagnetic shielding

## Abstract

Poly(lactic) acid nanocomposites filled with graphene nanoplatelets (GNPs) and multiwall carbon nanotubes (MWCNTs) are studied, varying the filler size, shape, and content within 1.5–12 wt.%. The effects of the intrinsic characteristics of nanofillers and structural organization of nanocomposites on mechanical, electrical, thermal, and electromagnetic properties enhancement are investigated. Three essential rheological parameters are identified, which determine rheology–structure–property relations in nanocomposites: the degree of dispersion, percolation threshold, and interfacial interactions. Above the percolation threshold, depending on the degree of dispersion, three structural organizations are observed in nanocomposites: homogeneous network (MWCNTs), segregated network (MWCNTs), and aggregated structure (GNPs). The rheological and structural parameters depend strongly on the type, size, shape, specific surface area, and functionalization of the fillers. Consequently, the homogeneous and segregated network structures resulted in a significant enhancement of tensile mechanical properties and a very low electrical percolation threshold, in contrast to the aggregated structure. The high filler density in the polymer and the low number of graphite walls in MWCNTs are found to be determinant for the remarkable shielding efficiency (close to 100%) of nanocomposites. Moreover, the 2D shaped GNPs predominantly enhance the thermal conductivity compared to the 1D shaped MWCNTs. The proposed essential structural parameters may be successfully used for the design of polymer nanocomposites with enhanced multifunctional properties for 3D printing applications.

## 1. Introduction

Polymer nanocomposites reinforced with graphene and carbon nanotubes represent a rapidly developing area of materials science in recent years due to novel functionalities and enhanced mechanical reinforcement of engineering and natural polymers, which greatly expands the ranges of their application [1,2]. The reinforcement of nanocomposites is usually related to the enormous surface area of the nanofillers and the strong interfacial interactions, ensuring an efficient transmission of load and functionalities between the filler and the polymer [2,3]. In order to obtain novel properties, the type, size, and shape of carbon nanofillers are usually varied when added in the polymer [4,5]. However, researchers reported contradictory results for nanocomposite materials depending on those factors in different polymers [4,5,6]. The main explanation for the essential physical mechanism of improved properties of nanocomposites in the literature is usually related with good dispersion and strong network connectivity of nanofillers. However, researchers emphasized that, although graphene sheets are separated and homogeneously dispersed within the polymer matrix, the full potential of graphene in nanocomposites still has not been realized [6,7,8]. The behavior, physical properties, and other functions of nanocomposites obviously depend on many factors, but filler type, matrix polymer, and composite microstructures are the most essential. Relatively limited research has been conducted to understand the intrinsic structure–property relationship in polymer nanocomposites, hence wide disagreement has been observed in the effects of nanosized reinforcements on nanocomposite properties [8]. Systematic studies are needed to investigate and understand the effect of reinforcement size and volume fraction of nanofiller for the enhanced multifunctionality of polymer nanocomposites with graphene and carbon nanotubes [5,8].

Recently, graphene and carbon nanotubes have been studied as additives in polymers in order to obtain multifunctional materials for application in additive manufacturing [1,7]. It should be highlighted that 3D printing is not only an innovative processing technology, but it is the future of the manufacturing industries. Therefore, unlimited needs exist for novel materials suitable for 3D printing for variety of applications that require improved mechanical performances, conductivity and other functional properties of the final products [9,10]. Due to the complexity of composition and processing, however, key issues for the mass production of carbonaceous nanocomposite materials for 3D printing include the selection of filler and polymer, control of the dispersion process, and interfacial interaction phenomena in order to obtain superior properties [2,4,5,6,7].

In our previous studies [10,11,12,13,14,15], we have reported on the development of innovative PLA-based nanocomposites incorporating graphene nanoplatelets and multiwall carbon nanotubes, suitable for 3D printing applications. The present study aims to identify the essential structural parameters for the design of enhanced mechanical, electrical, electromagnetic, and thermal properties of nanocomposites, compared to the neat PLA. The effects of intrinsic characteristics of carbon nanofillers (type, size, shape, aspect ratio, surface area, functionalization, etc.) on the rheological properties and microstructure of nanocomposites are investigated and related to the mechanical reinforcement, electrical and thermal conductivity, and electromagnetic shielding efficiency. The degree of dispersion, percolation threshold, interfacial interactions, and microstructural features are estimated as the essential nanostructure parameters which control the reinforcement and multifunctionality of nanocomposites.

## 2. Materials and Methods

### 2.1. Materials

In the present study, the adopted polymer is Ingeo™ Biopolymer PLA-3D850 (Nature Works, USA), suitable for 3D printing application, with MFR 7–9 g/10 min (210 °C, 2.16 kg), peak melt temperature 180 °C, and glass transition temperature 60–65 °C. Two types of graphene nanoplatelets (GNPs) and multiwall carbon nanotubes (MWCNTs) are selected. The GNP-a nanoplatelets are polydispersed (0.5–25 µm) with average lateral dimension (5–7 µm) and smaller aspect ratio (AR = 240) compared to GNP-b (5–10 µm; AR = 500). Meanwhile, the MWCNT-b has a much smaller diameter, length, and aspect ratio, but higher specific surface area (200–300 m^2^/g), compared to the MWCNT-a (110 m^2^/g). The MWCNTs are surface functionalized by -OH groups (MWCNT-a) and by oxidation (MWCNT-b), while GNPs are non-functionalized materials, according to the producer and own investigations [11]. The details on basic characteristics of fillers, trade name, and producers are summarized in Table 1.

Two series of nanocomposite formulation, NC-a and NC-b were fabricated through melt extrusion technique with varying filler contents from 1.5 to 12 wt.% (NC-a) and 1.5–9 wt.% (NC-b). The two series of nanocomposites differ in characteristics of the nanofillers, thus NC-a contains GNP-a and MWCNT-a, while NC-b contains GNP-b and MWCNT-b fillers. Firstly, masterbatches of 12 wt.% and 9 wt.% GNPs and MWCNTs were produced by melt mixing for the series NC-a and NC-b, respectively, using the twin screw extruder, COLLIN Teach-Line ZK25T, with a screw speed of 40 rpm and temperature of 170–180 °C. Then, the formulations with lower filler contents of 1.5, 3, 6 and 9 wt.% were prepared by dilution of masterbatches with the neat PLA through the second extrusion run. The composite pellets of all formulations were further extruded by a single screw extruder (Friend Machinery Co., China) for producing filament for 3D printing (FDM) with diameter 1.75 mm. The short processing time, the low processing temperatures and the addition of carbon nanofillers prevent the PLA degradation. Therefore, the decomposition processes are insufficient during the extrusion of nanocomposite filament, thereby insufficient effect of polymer degradation on rheological results is expected.

### 2.2. Experimental Methods

The rheological measurements were carried out with AR-G2 Rheometer (TA Instruments, New Castle, DE, USA) using electrical-heated parallel plate geometry of 25 mm diameter. Both oscillatory and shear flow test modes were applied, where the dynamic modulus (G’, G”) and the steady-state viscosity (η) were determined with varying frequency or shear rate, respectively, in the range 0.05–100 s^−1^, at temperature 200 °C. The test samples were hot-pressed disks of diameter 20 mm and thickness 2 mm. Advantage software was used for performing the experiments and rheological calculations.

Bright field transmission electron microscopy (TEM, FEI Company, Hillsboro, OR, USA) analysis was performed by using FEI TECNAI G12 Spirit-Twin (LaB6 source) instrument equipped with a FEI Eagle-4k CCD camera and operating with an acceleration voltage of 120 kV. Thin slides, cut from the cross section of the nanocomposite filaments at room temperature by ultramicrotome, were analyzed. The sections were placed on 400 mesh copper grids for testing.

Wide angle X-ray diffraction (WAXD) was employed for characterization of the crystalline structures. The measurements were performed by DX-1000 X-ray diffractometer (Dandong Fangyuan Instrument Co., Ltd., Dandong, Liaoning Province, China) employing Copper Line Focus X-ray tube producing Kα radiation by a generator operating at 45 kV and 25 mA. The diffraction patterns were undertaken in the range from 5° to 70° at a rate of 1°/min. The specimens been analyzed are powders for the fillers, GNPs and MWCNTs, as well as hot-pressed thin films, for the nanocomposites. The data from WAXD were processed using the MDI Jade software capable for analyzing the crystal structure of materials.

Tensile mechanical measurements were carried out using a Universal Mechanical Tester (UMT-2, Bruker, Billerica, MA, USA). The test specimens were filaments with circular cross-section of diameter 1.75 mm and length 50 mm. Tensile test was performed at room temperature by using 1 kN sensor, with a tensile speed of 1 mm/min. The standard error was determined by testing 10 samples of each composition.

A pico-ammeter (Keithley 2400, Keithley Instruments Inc., Beaverton, OR, USA) was used to measure the bulk conductivity. Three disc samples with a diameter of 16 mm and thickness 3 mm of each composition were tested, as prepared by hot-pressing at 180 °C. The test was conducted at room temperature of 20 °C with direct reading of the flushing current between two adjacent metallized zones. During the test, the electrical resistivity of the material in Ohm was measured. Electrical conductivity, σ [S/m], of the bulk sample was calculated using the equation: σ = *L*/*Rπr*^2^, where *R* is electrical resistivity, *L* is sample thickness, and *r* is radius of the sample.

Thermal conductivity was measured by a thermal constant analyzer (Hot Disk 2500, Hot Disk Inc., Göteborg, Sweden) through a transient plane source method. The measurements were performed with hot-pressed specimens of sizes 16 mm diameter and 3 mm thickness by putting the sensor (3 mm diameter) between two similar slabs of material. The sensor supplied a heat pulse of 0.01 W for 40 s to the sample at room temperature and the associated temperature changes were recorded. Three twin samples were tested for each nanocomposite formulation.

The electromagnetic response of the samples was evaluated in terms of scattering parameters. The test samples were films with thickness of 1 mm, prepared by hot-pressing. The transmitted/input (S_21_) and reflected/input (S_11_) signals are investigated in the Ka-band frequency range (from 26 GHz to 37 GHz) by using a scalar network analyzer R2-408R (ELMIKA, Vilnius, Lithuania), equipped with 7.2 × 3.4 mm waveguide system. The reflection (*R*), transmission (*T*), and absorption (*A*) coefficients are derived from the measured S-parameters, such as: $R=S_{11}^{2},\;T=S_{21}^{2},\;A=1-R-T.$ Electromagnetic shielding efficiency was computed as a sum of absorption and reflection (*SE* = *A* + *R*, in %).

## 3. Results

### 3.1. Essential Rheological Parameters

The rheological characteristics of the NC-a (with GNP-a, MWCNT-a) reported previously [12,13,14,15] are compared to those of the NC-b nanocomposites, loaded with GNP-b, MWCNT-b, with varying filler contents of 1.5–9 wt.%, in the PLA matrix polymer. Three essential rheological parameters, which determine the rheology–structure relationship in nanocomposites are identified, as: (i) the degree of dispersion, (ii) the percolation threshold, and (iii) the interfacial interactions.

#### 3.1.1. Degree of Dispersion

The degree of dispersion of nanocomposites is evaluated by the “flow index” (n), which is determined by the fitting of experimental viscosity data at low shear rates ($\dot{\gamma }$→0 s^−1^) with the power law fluid model [4],
(1)$$\eta =\dot{K}{\dot{\gamma }}^{n-1}$$ where η is the steady-state viscosity, $\dot{\gamma }$ is the shear rate, *K*—the flow consistency index at ($\dot{\gamma }$→0). Power law slope is (*n* − 1), where (*n*) is the “flow index”. For a fixed filler type and filler content, the low flow index (*n*) is associated with the high degree of dispersion. Theoretically, the flow index values of *n* ≤ 0.5 corresponds to liquid-to-solid transition, which is related to percolation.

The degree of dispersion, estimated by the flow index (*n*) is demonstrated in Figure 1, where the steady-state viscosity vs. shear rate of the two nanocomposite series, NC-a and NC-b are compared for two filler contents, 1.5 wt.% (Figure 1a) and 6 wt.% (Figure 1b), respectively. The flow index (n), as calculated by Equation (1), as well as the initial viscosity values, at $\dot{\gamma }$ = 0.05 s^−1^ are summarized in Table 2. In the case of the neat PLA and the 1.5wt.% GNP composites, the power law slope is of (n-1)→0, which determine the flow index of (n→1), therefore a Newtonian-like behavior. Meanwhile, for 1.5 wt.% MWCNT composites, as well as for all composites with 6 wt.% filler content, a Bingham-like behavior is observed, having a power law slope of Equation (1), (*n* − 1) >> 0, so the flow index is of (*n*) < 0.5, as shown in Table 2.

A different flow behavior is observed for the MWCNT and the GNP based nanocomposites. In general, the MWCNT-based nanocomposites demonstrate very low flow index (*n* < 0.5) and high initial viscosity, even at 1.5 wt.%, which is associated with the high degree of dispersion of MWCNTs in the PLA matrix. This is obviously due to the surface functionalization and the high specific surface area of nanotubes. In contrast, the GNP nanocomposites show a high flow index (*n* > 0.5), even at 6 wt.% filler content, which may be associated with low degree of dispersion, due to the absence of functionalization and low specific surface area of the GNPs.

Moreover, if compare the fillers of the same type, the MWCNT-b nanocomposites shows twice lower flow index and a decade higher initial viscosity, compared to the MWCNT-a, and therefore twice higher degree of dispersion of MWCNT-b than the MWCNT-a in the PLA matrix. This is obviously due to much higher specific surface area of MWCNT-b (250–300 m^2^/g) compared to MWCNT-a (110 m^2^/g). Concerning GNPs, the flow index and the initial viscosity of GNP-a and GNP-b nanocomposites are quite similar, so both fillers have a similar degree of dispersion in the PLA matrix. It is a result of identical specific surface area (30–40 m^2^/g) of both non-functionalized GNP fillers.

#### 3.1.2. Rheological Percolation Threshold

The rheological percolation threshold (RPT) is associated with the transition from Newtonian to plastic flow behavior, due to the formation of continuous network of interconnected nanoparticles, which built uniform agglomerates, so called fractals [12,15,16]. In Figure 2a, the relative viscosity at a low shear rate ($\dot{\gamma }$ = 0.05 s^−1^) vs. filler content is presented, comparing the MWCNT and the GNP nanocomposites from the two series, NC-a and NC-b. The relative viscosity, $\eta_{rel}$ = $\eta /\eta_{o},$ increased nonlinearly with increasing filler content, where $\eta_{rel}$ values for the MWCNT nanocomposites are of 1–2 decades higher compared to the GNPs. At low filler contents, the viscosity function vs. filler content fit well with the adapted Einstein type equation [13,17]: (2)$$\eta_{rel}=1+\left[\eta \right]\varphi ,$$ where *φ* is the filler fraction, [*η*] is the intrinsic viscosity depending on the size, shape, and aspect ratio of the filler, shown in Table 2. In Figure 2a the continuous lines present the B-spline curve-fitting of the relative viscosity data at a fixed shear rate $(\dot{\gamma }=0.05\;s^{-1}$), while the dotted lines show the fitting by the adapted Einstein type linear equation, Equation (2). The deviation of the relative viscosity function from the linear Equation (2) is used to determine the critical filler concentration, named rheological percolation threshold, (φ_p_).

Another criterion for determination of the RPT is based on the fractal model (scaling law), used for colloidal dispersions [16,18]. The percolation threshold is determined from the starting point of the scaling law. (3)$$G_{o}^{\prime }\;\sim \;\varphi^{m},$$ where $G_{o}^{\prime }$ is the terminal storage modulus at low frequencies *ω*→0 rad/s, *φ* is the fraction of the filler, and *m* is the scaling exponent, which depends on the fractal dimension, where m = 5/(3 − D), with coefficient D representing the fractal size. Figure 2b presents the plateau storage modulus vs. filer fraction.

In Figure 2a,b the two nanocomposite series NC-a (with GNP-a, MWCNT-a) and NC-b (with GNP-b, MWCNT-b) are compared and the experimental data are fitted with Equations (2) and (3), respectively, for estimation of the rheological percolation threshold. From experimental data and theoretical calculations, the RPT is determined as φ_p_ ~ 5 wt.% for GNP-a and φ_p_ ~ 6 wt.% for GNP-b nanocomposites, respectively. While much lower RPT values are obtained for the MWCNT-b (φ_p_ ~ 0.5 wt.%) and MWCNT-a (φ_p_ ~ 1.5 wt.%), as shown in Table 2.

The rheological percolation threshold of nanocomposites, NC-a and NC-b is related with the formation of fractals of nanoparticles in the PLA matrix, resulting in solid-like behavior. The fractals are uniform agglomerates of similar size and structure that are formed by the GNPs and MWCNTs particles, respectively, and the immobilized polymer matrix in between. The most pronounced parameters which determine the low percolation threshold of MWCNT-b are obviously associated with higher degree of dispersion and a twice higher specific surface area, as well as nearly a decade lower aspect ratio, compared to the MWCNT-a. In contrast, the degree of dispersion is similar for GNP-a and GNP-b nanocomposites, thus the slightly lower percolation threshold of GNP-a is obviously due to the twice lower aspect ratio, compared to that of GNP-b.

For the NC-a and the NC-b nanocomposites, the scaling exponent (m), determined by Equation (3) is used for calculation of the fractal dimensions D and the values are summarized in Table 2. Above the percolation threshold, the rigid and platelet shaped GNP-a and GNP-b particles with similar degree of dispersion are organized in fractals of similar size, D = 1.861 and D = 1.896, respectively. In contrast, the highly dispersed, entangled, and needle-like MWCNT-b and MWCNT-a fillers formed smaller size fractals (D = 1.195 and D = 1.311, respectively), compared to those of GNPs.

#### 3.1.3. Interfacial Interactions

The interfacial polymer-filler interactions significantly affect the relaxation processes in polymer nanocomposites [19]. The presence of nanofiller particles and the lack of relaxation of the polymer chains contribute to the solid-like response and the non-terminal behavior at low frequencies. Therefore, it is important to estimate the relaxation processes in polymer systems by calculation of the relaxation-time spectra using linear viscoelastic data from experimental rheology and constitutive equation models [4,19]. Researchers reported that the interface phenomena might cause local or global changes in the polymer relaxation associated with the strength of the interfacial polymer-filler interactions [4,20,21]. Herewith, the relaxation-time spectrum, H(τ) was calculated by fitting the generalized Maxwell-type model, Equations (4) and (5), to data of the experimental dynamic moduli by using Advantage software (TA Instruments). (4)$$G^{\prime }\left(\omega \right)=G_{e}+\int_{-\infty }^{\infty }H\left(\tau \right)\frac{\omega^{2}\tau^{2}}{1+\omega^{2}\tau^{2}}dln\tau$$
(5)$$G^{\prime \prime }\left(\omega \right)=\int_{-\infty }^{\infty }H\left(\tau \right)\frac{\omega \tau }{1+\omega^{2}\tau^{2}}dln\tau$$ where *τ* is relaxation time, H(τ) is relaxation spectrum, $G^{\prime }\left(\omega \right)$ is storage modulus, $G^{\prime \prime }\left(\omega \right)$ is loss modulus, and $G_{e}$ is equilibrium plateau modulus.

Figure 3 compares the example relaxation-time spectra for the neat PLA and nanocomposites of 6 wt.% filler content, containing GNP-a, GNP-b, MWCNT-a, and MWCNT-b, respectively. As seen, the relaxation spectra of nanocomposites is shifted towards longer relaxation time, as well as to higher weight values, H(τ), compared to the spectra of the neat PLA. This type of prevented relaxation corresponds well to the solid-like behavior of the nanocomposites around and above the percolation threshold.

For illustration of this effect, the weight of the spectra, H(τ) for an example fixed relaxation time, τ = 50 s is summarized for the PLA, the NC-a and NC-b series in Table 2. Specifically, at this fixed relaxation time, the weight of the spectra for the MWCNT-b nanocomposite is shifted about 9 decades towards higher values, compared to the neat PLA. This effect become a decade lower for the MWCNT-a nanocomposite. Much lower shift of the spectra in vertical position is observed for GNP-a (5 decades) followed by GNP-b (3 decades) compared to the neat PLA. Moreover, the MWCNT-b, followed by MWCNT-a and GNP-a fillers have a strong effect on broadening of the spectra of 6 wt.% nanocomposites towards infinite relaxation times. At the same time, GNP-b shifts the spectra only by a decade to longer relaxation time, compared to the spectra of the neat PLA. This may be associated with the different level of altering the mobility of polymer segments by nanofillers. Thus, the MWCNT-b and MWCNT-a interfaces evidently causes global changes in the polymer relaxation due to the large specific surface area and the high degree of dispersion. Meanwhile, the GNP-a and GNP-b interfaces cause a predominantly local energetic barrier for the segmental motion of the polymer chains.

### 3.2. Microstructure and Morphology of Nanocomposites

#### 3.2.1. XRD Analysis of Materials

XRD analysis is applied for the characterization of nanocomposites, as it is a very sensitive analysis of the crystal structure of materials. In Figure 4a, the WAXD diffraction patterns are summarized for the fillers, GNPs and MWCNTs. The pure MWCNT-а exhibits a basal reflection (002) peak at 2θ = 25.8°, while for MWCNT-b, 2θ = 25.2°, corresponding to the d spacing between graphitic walls of 0.34 nm for MWCNT-a and 0.35 nm for MWCNT-b, respectively. Moreover, the spectral profile of MWCNT shows a weak and broad XRD peak at 2θ = 43°, which is assigned to graphitic (100) crystalline lattice. In contrast, the pristine GNP-a and GNP-b show a sharp, intensive peak at 2θ = 26.35° indicating a more crystalline structure for GNPs, compared to that of MWCNTs. The weak peak at 2θ = 54.4° in the GNP spectra is assigned to graphitic (004) crystalline lattice. In contrast, a broad amorphous halo was observed around 2θ = 15.5° in the diffraction spectra of the neat PLA, this indicating that the biodegradable polymer PLA has a predominantly amorphous microstructure.

The analysis of the patterns of 6 wt.% nanocomposites in Figure 4b, the GNP-a and GNP-b nanocomposites shows that a distinct diffraction is observed at 26.3° relating to an interlayer spacing of 0.34 nm, based on the Bragg’s law, which is associated with the graphitic (002) plane. No changes in the position of the peak corresponding to basal reflection (002) of GNPs is observed, therefore the spacing between graphene nanoplatelets is not changed in the nanocomposites during processing. The high intensity recorded at 6 wt.% GNPs loading could be attributed to a relatively large number of graphene layers organized in stacks in the nanocomposites. This confirm the low degree of dispersion and insufficient exfoliation of the GNP nanoplatelets in PLA matrix.

In the carbon nanotube composites, graphitic (002) pattern is slightly visible for the 6 wt.% MWCNT-a, but it is invisible for the 6 wt.% MWCNT-b nanocomposite. The disappearance of the peak (002) can be attributed to the low dimensionality of the MWCNT agglomerates and good dispersion due to functionalization of the carbon nanotubes. The WAXD results confirm the rheological results in Section 3.1 for a higher degree of dispersion of MWCNT-b fillers followed by MWCNT-a in the PLA matrix, compared to GNPs.

The WAXD technique and MDI Jade software were used to calculate the stacking thickness of the GNP crystallites by applying the Scherer equation. It was found that powders GNP-a and GNP-b have a stack thickness of about 18 nm and 22 nm, respectively, which is similar to that in the technical specifications. The stack dimensions slightly increase in nanocomposites having 6 wt.% GNPs (34 nm for GNP-a and 29 nm for GNP-b, respectively), which is associated with a tendency to agglomeration. The addition of GNPs and MWCNTs is not affected the amorphous broad halo, corresponding to the PLA, therefore the nucleation effect of both fillers is insufficient.

#### 3.2.2. TEM Morphological Analysis

Morphological analysis of nanocomposites is performed by transmission electron microscopy (TEM), in order to visualize the degree of dispersion and structural organization of the fillers in the PLA matrix, with varying the filler type. The example micrographs of nanocomposites at 6 wt.% filler content are compared in Figure 5, where the morphology of nanocomposites around the percolation threshold (φ_p_ ≤ 6 wt.%) for GNPs and above the percolation threshold for MWCNTs (φ_p_ ≤ 1.5 wt.%) is shown. The nanocomposite series NC-a and NC-b are compared in Figure 5, where the first line presents the micrographs of GNP-a and GNP-b composites, while the second line shows the MWCNT-a and MWCNT-b, respectively. All images are compared at the same magnification. As seen, the state of dispersion and the filler distribution at 6 wt.% filler content depend strongly on the type of the filler. The GNPs demonstrate aggregated structure with broken continuity consisting of large graphene particles of size, from 500 nm to 2 µm. The GNP-a particles seems to be partly exfoliated, while GNP-b particles are only dispersed in thick stacks. These results confirm the rheological findings and the WAXD results above.

In contrast, in the second line of Figure 5, the 6 wt.% MWCNT-a nanocomposite shows a segregated network structure of interconnected carbon nanotubes or their small agglomerates, which form continuous pathways in the polymer matrix [7,22]. The segregated network structure is composed of regions rich and poor in nanofillers. In contrast, the MWCNT-b nanocomposite has a homogeneous network structure of well-dispersed entangled nanotubes. Obviously, the smaller size of MWCNT-b (diameter 9.5 nm, length 1.5 µm) produced larger number of dispersed nanotubes, i.e., high filler density in the PLA matrix, compared to the MWCNT-a (diameter < 30 nm, length 10–30 µm).

Therefore, we may conclude that three structural types are formed in the NC-a and NC-b composites above the percolation threshold, by increasing the degree of dispersion, i.e., the filler density in the PLA matrix: aggregated structure (GNP-a and GNP-b), segregated network (MWCNT-a), and homogeneous network (MWCNT-b). The effects of these three structural types on mechanical, electrical, electromagnetic and thermal properties of nanocomposites will be analyzed in the next sections.

### 3.3. Mechanical Properties

Considering the influence of MWCNTs and GNPs fillers on the mechanical properties of PLA nanocomposites, the tensile mechanical characteristics of the filament are analyzed, such as tensile ultimate strength, elongation at ultimate strength and Young’s modulus. The results are summarized in Table 3. The dependence of Young’s modulus vs. filler contents of NC-a and NC-b nanocomposite series with GNPs and MWCNTs is shown in Figure 6.

As seen in Figure 6a for NC-a series, the Young’s modulus of the GNP-a filler is almost unchanged by increasing the filler content, compared to the neat PLA. However, the ultimate strength and elongation decreased significantly by increasing the GNP-a content, thus ~25% lower values are observed at 12 wt.% GNP-a, compared to the neat PLA (Table 3). In contrast, for the MWCNT-a composites, the Young’s modulus gradually increases by increasing the filer content and maximal improvement (~20%) is achieved at 12 wt.% MWCNT-a. Moreover, the ultimate strength and elongation of the MWCNT-a nanocomposites are similar to that of the neat PLA, and slightly decreased only for 12 wt.% filler content.

For NC-b series, more significant effects of the fillers on the mechanical properties of nanocomposites are observed. In Figure 6b, the Young’s modulus of the GNP-b composite increases gradually, with ~12% at 6 wt.% filler content (the RPT), while at 9 wt.%, it decreases, in comparison with the neat PLA. The ultimate strength and elongation remain unchanged by increasing the GNP-b content keeping values similar to that of the neat PLA. In contrast, the MWCNT-b nanocomposites demonstrate the highest improvement (~30%) of Young’s modulus by increasing the filler content up to 6–9 wt.%. Moreover, an increase of ultimate strength by 32% and of elongation by 12% is observed for the 6 wt.% composites, compared to the neat PLA. (Table 3). A further increase of the MWCNT-b contents to 9 wt.% resulted in a decrease of strength and elongation.

By correlation of the nanocomposites’ mechanical properties with the essential structural parameters, we conclude that the homogeneous network structure with highest filler density, formed in the MWCNT-b nanocomposites around ~ 6 wt.% filler content, is the most promising to achieve strong enhancement of tensile mechanical characteristics, such as Young’s modulus, tensile strength, and elongation. The segregated network structure of the MWCNT-a nanocomposites results in a slight improvement of the tensile characteristics. Meanwhile, the aggregated structure of GNP-a and GNP-b nanocomposites leads to almost unchanged Young’s modulus by increasing the filler content, however the ultimate strength and elongation are significantly compromised if the aspect ratio of GNPs is lower than 500. Obviously, the homogeneous network structure with high filler density, followed by the segregated network structure are preferable for MWCNT nanocomposites, for obtaining tensile properties enhancement, as they carry high levels of transferring stress across filler-polymer interfaces [22,23,24], in contrast to the aggregated structure of GNP nanocomposites.

It may also be assumed that the high degree of dispersion of the MWCNT-b, followed by MWCNT-a filler, determined as well by rheology, allow to express the unique mechanical strength of the individual carbon nanotubes and to transfer load to the polymer [22,24]. However, at high filler contents, e.g., 9 and 12 wt.%, the homogeneity of MWCNT dispersion could deteriorate, and this could destroy the homogeneity of the network structure, resulting in a decrease of tensile strength and elongation.

### 3.4. Electrical Conductivity

The variation of the bulk conductivity of PLA nanocomposites versus the concentration of MWCNTs and GNPs fillers was reported in the previous studies [14,15,22,25,26]. It was stated, that with increasing the filler content, the carbonaceous nanoparticles began to form a three-dimensional conductive network within the polymer, resulting in a sharp increase of the overall electrical conductivity, which was associated with electrical percolation threshold (EPT). In fact, below the EPT, the composites are non-conductive, similar to the neat PLA.

The electrical conductivity above the percolation region is usually described by the following power law equation:(6)$$\sigma =\sigma_{o}{\left(\varphi -\varphi_{p}\right)}^{t}$$ where $\sigma_{o}$ is the conductivity of the filler, φ is the filler content, $\varphi_{p}$ is the EPT, and t is a critical exponent linked to the morphological arrangement of the filler in the percolating structure.

Figure 7 illustrates the percolation curves of electrical conductivity vs. filler content, comparing the nanocomposites with different filler characteristics: MWCNT-a, MWCNT-b, GNP-a, and GNP-b, respectively. The test samples are hot-pressed disks. Consistent with percolation theory, as the filler concentration increases above a critical filler content (i.e., EPT), the electrical conductivity increases sharply to several orders of magnitude compared to the unfilled PLA. This phenomenon indicates that electrically conductive paths between interconnected conductive nanoparticles are established within the nanocomposite, thus inverting its behavior from insulating to conductive, since the electrons can flow due to the tunneling effect.

Based on the experimental data from Figure 7 and the power law Equation (6), the EPT is calculated of 0.5 wt.% for the MWCNT-b and 1.4 wt.% for the MWCNT-a nanocomposites. While, the estimated EPT is 5 wt.% for the GNP-a nanocomposite and 6 wt.% for the GNP-b, respectively (Table 2). In general, it may be considered, that the electrical percolation threshold coincide well with the rheological percolation threshold (EPT ≈ RPT), thereby similar parameters would determine both characteristics. Thus, for the MWCNT-b nanocomposites, the homogeneous network structure with high filer density, high degree of dispersion, high specific surface area, and strong interfacial interactions are obviously the essential structural parameters, which determine the very low EPT. The segregated network structure of the MWCNT-a nanocomposites with lower specific surface area of the filler and weaker interfacial interactions than the MWCNT-b one show slightly higher EPT value. In contrast, the aggregated structure of GNP-a and GNP-b nanocomposites with quite low degree of dispersion and weak interfacial interactions produce high EPT values. These structural parameters also determine the high dc-conductivity at high filler content, e.g., at 9 wt.% filler the following values are achieved: 1.5 S/m and 0.3 S/m for PLA reinforced with MWCNT-b and MWCNT-a, as well as 0.08 S/m and 2.7 × 10^−5^ S/m for nanocomposites with GNP-a and GNP-b, respectively.

### 3.5. Thermal Conductivity

The GNPs and MWCNTs have attracted great attention for their exceptional intrinsic ability to conduct heat with theoretical values of 5000 (W/mK) and 3000 (W/mK), respectively [10,27]. Moreover, thermal conductivity through a polymer is a complex process, influenced by many parameters, such as crystallinity, orientation of the macromolecules, etc., where phonons are usually considered as thermal carriers in polymers [25]. In our previous studies, GNP-a and MWCNT-a were used as fillers to enhance the thermal conductivity of PLA composites with varying filler contents [10,25,26]. Both 3D printed and hot-pressed test samples were studied. The thermal conductivity of nanocomposites was found to increase almost linearly with increasing of the filler content. Much higher improvement of thermal conductivity by GNP-a filler was found than that of MWCNT-a.

In the present study, Figure 8 presents the thermal conductivity vs. filler content of the hot-pressed nanocomposite samples comparing the effects of different types of fillers, GNP-a, GNP-b, MWCNT-a and MWCNT-b, with varying filler contents within 1.5–12 wt.%. In general, three essential structural parameters are observed which govern the thermal conductivity of nanocomposites with GNP and MWCNTs, such as filler shape, degree of dispersion, and interfacial interactions: (i)2D vs. 1D shape of the filler: The nanocomposites containing 2D-predominant shape fillers, such as graphene nanoparticles (GNPs) showed much better heat conduction compared to the 1D-nanotubes (MWCNTs). Thus, the total surface wetting of 2D-GNP nanoplatelets by PLA polymer favors the strong binding among the graphene surfaces and the PLA matrix, thereby a well-established morphological network of filler within the composite. Therefore, the interface contacts between 2D (GNPs) and matrix polymer are characterized by much lower thermal Kapitza resistance (Rk), if compared with the 1D (MWCNTs), where the inner surfaces of the nanotubes are poorly wetted by the PLA. This means a more effective phonon heat flow in 2D-nanoplatelets, rather than 1D-nanotubes, that matched well to those in organic molecules, thus lowering the differences of phonon density of states between the two phases [10,26].(ii)Degree of dispersion and interfacial interactions. Better dispersion and stronger interfacial interaction of both 2D and 1D fillers in the matrix polymer resulted in higher values of thermal conductivity, due to stronger suppressing of phonon scattering. In particular, at a fixed filler content (i.e., 9 wt.%), the higher interfacial interactions in the GNP-a nanocomposites resulted in 254% improvement of thermal conductivity, while 223% for the GNP-b, compared to the neat PLA. In the case of 1D-MWCNTs, the higher degree of dispersion and stronger interfacial interactions of MWCNT-b resulted in a 113% increase in thermal conductivity, compared to a 94% increase for the MWCNT-a, with respect to the matrix PLA.

### 3.6. Electromagnetic Shielding

When an electromagnetic wave impacts on a surface, a part of it is absorbed by the material, another part is reflected, and the rest is transmitted through the material, in agreement with the following power balance: A + R + T = 100%, where A, R, and T are the absorption, reflection, and transmission power counterparts, respectively [28,29,30]. In our previous studies, it was found that the inclusion in the PLA matrix of the conductive fillers, such as MWCNT-a and GNP-a, in concentrations around and above the percolation threshold can attenuate the penetrating wave thanks to the effective conductive network, established in the material [10,13,31]. Recently, researchers reported for polyethylene composites with segregated carbon nanotubes network, which show high electromagnetic interference shielding efficiency [7,32].

In the present section, we consider the electromagnetic (EM) properties of 1 mm thick PLA-based nanocomposite films, NC-a and NC-b series, with varying contents of GNPs and MWCNTs, around and above the percolation threshold. In particular, their electromagnetic properties were investigated in the microwave spectrum (26–37 GHz) also referred as Ka-band. Figure 9 presents the reflection (R), absorption (A) and shielding efficiency (SE) vs. filler content, at 30 GHz, where Figure 9a shows the NC-a series with GNP-a and MWCNT-a, while Figure 9b compares the NC-b nanocomposite series, with GNP-b and MWCNT-b fillers, respectively.

For the neat PLA, the values of A, R and SE are close to zero at 30 GHz, thus confirming PLA to be an EM-waves transparent material. In contrast, in Figure 9a, the NC-a series with GNP-a and MWCNT-a, demonstrate remarkable electromagnetic properties. The shielding efficiency sharply increases from zero for PLA to SE ~ 60%, if only 1.5 wt.% of the two conductive fillers are added. By increasing the filler content up to 12 wt.%, the shielding efficiency gradually increases to about 90%, where MWCNT-a shows slightly higher SE values than GNP-a. In general, the shielding efficiency of both nanocomposites mostly due to high reflection and partial absorption properties. The reflection properties of GNP-a are slightly higher than that of MWCNT-a in the entire concentration range, while, in contrast, the absorption properties of GNP-a are lower than that of MWCNT-a. For NC-b series in Figure 9b, the MWCNT-b nanocomposite material demonstrate remarkable shielding efficiency, SE = 93% at 1.5 wt.% filler content and it achieves full shielding, SE ~ 100% with increasing the filler content to 6–9 wt.%. However, the GNP-b nanocomposite demonstrates lower shielding efficiency, SE = 40–75% within 1.5–9 wt.% filler content range. Interestingly, the GNP-b shows up to 25% higher reflection and 10-times lower absorption properties than that of MWCNT-b.

Based on the above results, the following structural parameters may be estimated as essential for obtaining remarkable electromagnetic properties of MWCNT and GNP nanocomposites in the GHz region:(i)Degree of dispersion, filler density, and structural organization: As shown in Section 3.1 and Section 3.2 above, the MWCNT-b demonstrate highest degree of dispersion in PLA matrix polymer, thereby much higher filler density in a volume of polymer at a fixed filler concentration, compared to MWCNT-a. Hence, the 6 wt.% MWCNT-b nanocomposite formed a homogeneous microstructure, while the MWCNT-a build a segregated structure of nanotubes, with poor and enriched by MWCNT regions. These structural features may be associated with the maximum shielding efficiency, SE~100% for MWCNT-b nanocomposites followed by SE ~ 90% of MWCNT-a within 6–9 wt.% filler contents. In case of GNPs fillers, the shielding efficiency of 80–86% for GNP-a, and SE = 75–80% for GNP-b nanocomposites, due to the aggregated structure with low filler density and low degree of dispersion of both fillers. Recently, researchers also reported the low frequency plasmon and high electromagnetic interference shielding efficiency of composites with a segregated carbon nanotubes network [32].(ii)Number of graphitic walls/layers in nanoparticles and interfacial interactions: The MWCNT-b nanotubes contain smaller number of graphitic walls (~5), compared to the MWCNT-a (~20), as estimated by the differences between the outer and the inner diameter. Therefore, the remarkable electromagnetic shielding efficiency of 6–9 wt.% MWCNT-b nanocomposites (SE = 100%), compared to MWCNT-a (SE = 90%), may be associated with 4-fold smaller number of the graphitic walls in MWCNT-b. Along with the simple geometrical explanation (the thinner the tube within the same weight percentage, the larger the number of individual tubes forming a percolative network), this effect also has an electromagnetic origin. In the microwave range, for long MWCNTs there exists the electromagnetic screening, when wave could not penetrate fully inside the tube, and therefore not all walls take part in the EM interaction. This means that the use of MWCNT with as small as possible number of walls is preferable, in favor of MWCNT-b [33]. For GNP nanocomposites, the GNP-a and GNP-b fillers have a stack thickness of 34 nm and 29 nm, respectively, as shown by WAXD analysis in Section 3.2.1. Therefore, both GNP fillers have similar number of graphitic layers, which may explain the similar electromagnetic shielding efficiency of both nanocomposites, SE~80%. However, the GNP-b nanocomposites demonstrate high R = 75% and very low A = 5% at 9 wt.% filler content, in contrast to the GNP-a, with R ~ 60% and A ~ 20%. This may be due to the stronger interfacial interactions in GNP-a nanocomposites compared to GNP-b one, allowing easy transfer of functionality from filler to the polymer.

The EM interference of the studied composites is mostly determined by reflection. However, it worth to mention that high reflection ability of 1 mm thick hot-pressed PLA composites does not necessarily mean that 3D printed structure/device will be reflective. The response of device is determined by its geometry mostly and the conductivity of the skeleton it composed of. It was already shown that to archive perfect absorption for 3D printed matched loads, sandwich structures, and meshes, the intermediate conductivity must be at the level of 1–200 S/m [34,35].

The applicability of PLA-based nanocomposites filled with GNPs and MWCNTs was verified to FDM technology in our previous studies [10,11,12,13,14,15,22,23]. The printability window of the nanocomposite filament was identified to be similar to that of the neat PLA for the percolated nanocomposites, while at filler contents highly above the percolation threshold, the printability window is shifted towards higher printing rates. A comparison was made between the extruded filament, the 3D printed samples, and the hot pressed one [22] with respect to their electrical conductivity and tensile properties. It was found that the filament demonstrates 2–3 decades higher electrical conductivity and 50–150% higher tensile mechanical characteristics, compared to that of 3D printed and hot-pressed samples, due obviously to some preferential orientation of anisotropic carbon nanoparticles during the filament extrusion, which favor the percolation network. Moreover, the 3D printed samples demonstrate almost 20–50% higher crystallinity compared to the hot-pressed samples. However, the enhanced crystallinity has insufficient effect on the tensile properties. Our results also indicate that there is a close relationship between mechanical properties and morphological structure of nanocomposites, both deeply influenced by the type, the concentration and the degree of dispersion of nanofiller. Recently, researchers [36,37] reported that the deposition orientation has a significant influence on the mechanical behavior of the FDM 3D printed parts, compared to the filament. All these aspects may be associated with anisotropy of electrical and mechanical properties attributed to the nanofiller orientation in the matrix polymer and influenced by the deposition-induced effects during FDM. These effects of 3D printing will be investigated in more detail in our future studies.

## 4. Conclusions

In the present study, a nanocomposite series containing two types of GNP and MWCNT fillers of different intrinsic characteristics were investigated, varying the filler contents within 1.5–12 wt.%. Three essential rheological parameters of nanocomposites were identified which determine the rheology–structure relationship in nanocomposites: degree of dispersion, percolation threshold, and interfacial interactions. It was found that the rheological parameters depend strongly on the type, size, specific surface area, and functionalization of the fillers. Moreover, above the percolation threshold, the nanocomposites form three types of structural organization: aggregated structure (GNPs), segregated network and homogeneous network (MWCNTs), whit increasing the filler density and the degree of dispersion. The identified rheological and structural parameters are determinant for the strong enhancement of mechanical properties and electrical conductivity. The electrical percolation threshold was found to coincide well with the rheological percolation threshold. Importantly, the homogeneous network structure of MWCNT-a, followed by the segregated network structure of MWCNT-b nanocomposites resulted in a significant enhancement of tensile mechanical properties, in contrast to the aggregated structure of GNP-based nanocomposites. The high degree of dispersion, high filler density, and particularly, the low number of graphitic walls in the MWCNTs are found determinant for the remarkable shielding efficiency (SE = 100%) of the nanocomposites. Meanwhile, the thermal conductivity is predominantly enhanced by the 2D shape of GNPs compared to the 1D shaped MWCNTs. Importantly, the stronger interfacial interaction in the 2D filler GNP-a, resulted in higher values of thermal conductivity compared to GNP-b, due to a better suppressing the phonon scattering. The selected essential structural parameters of carbon nanotubes and graphene are successfully used for the design of multifunctional polymer nanocomposites and tailoring of their structure and properties for applications in the 3D printing of devices for sensing and electromagnetic shielding.

## Figures and Tables

**Figure 1 polymers-12-01208-f001:**
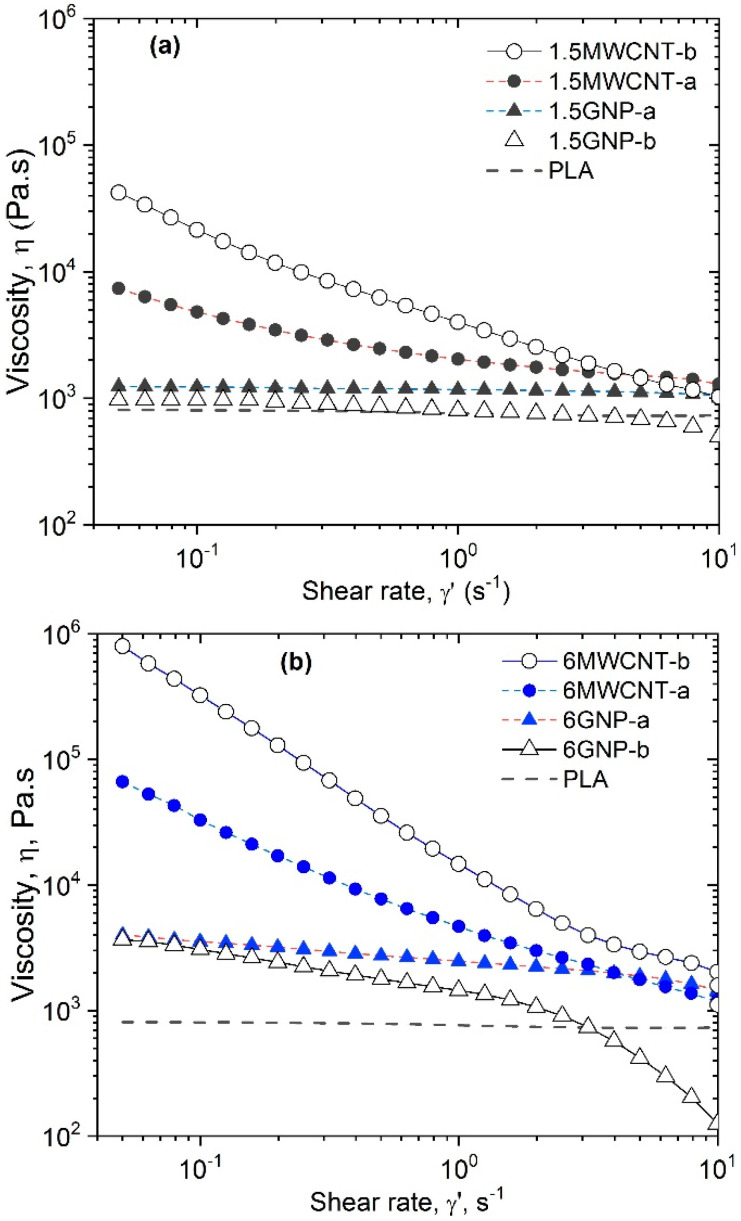
Viscosity vs. shear rate of NC-a and NC-b series comparing nanocomposites with MWCNTs and GNPs with the neat PLA for: (**a**) 1.5 wt.% and (**b**) 6 wt.% filler contents.

**Figure 2 polymers-12-01208-f002:**
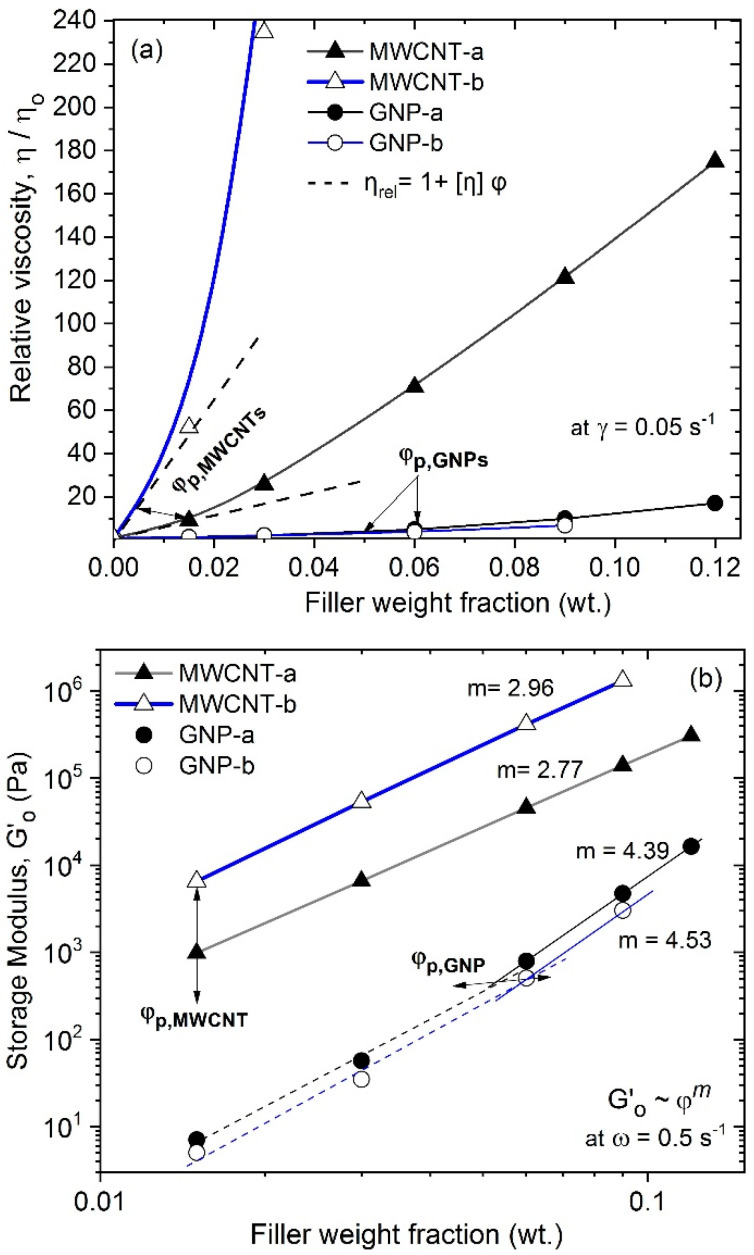
(**a**) Relative viscosity (at $\dot{\gamma }$ = 0.05 s^−1^) and (**b**) terminal storage modulus (at *ω* = 0.5 s^−1^) vs. filler content for nanocomposites with MWCNT and GNP, series NC-a and NC-b. Arrows show the percolation threshold.

**Figure 3 polymers-12-01208-f003:**
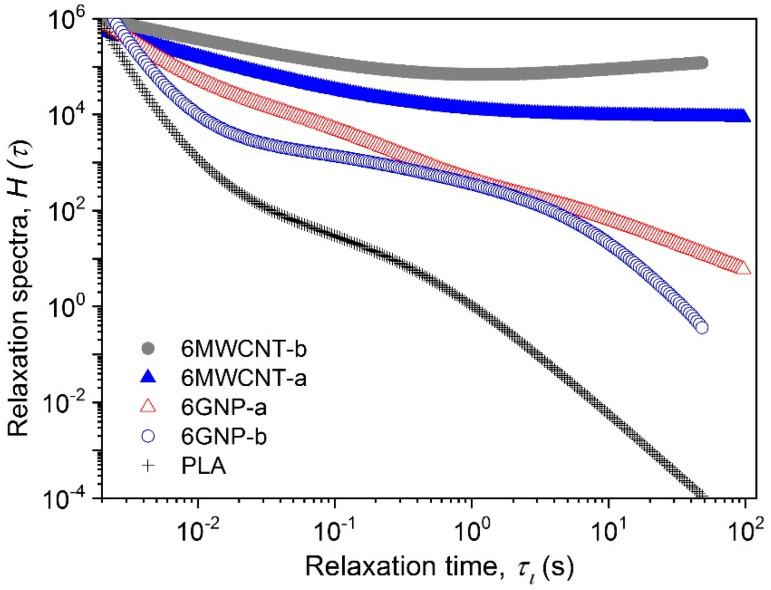
Relaxation time spectra of PLA nanocomposites with 6 wt.% filler content. Comparison of GNP-a, GNP-b, MWCNT-a and MWCNT-b nanocomposites with neat PLA.

**Figure 4 polymers-12-01208-f004:**
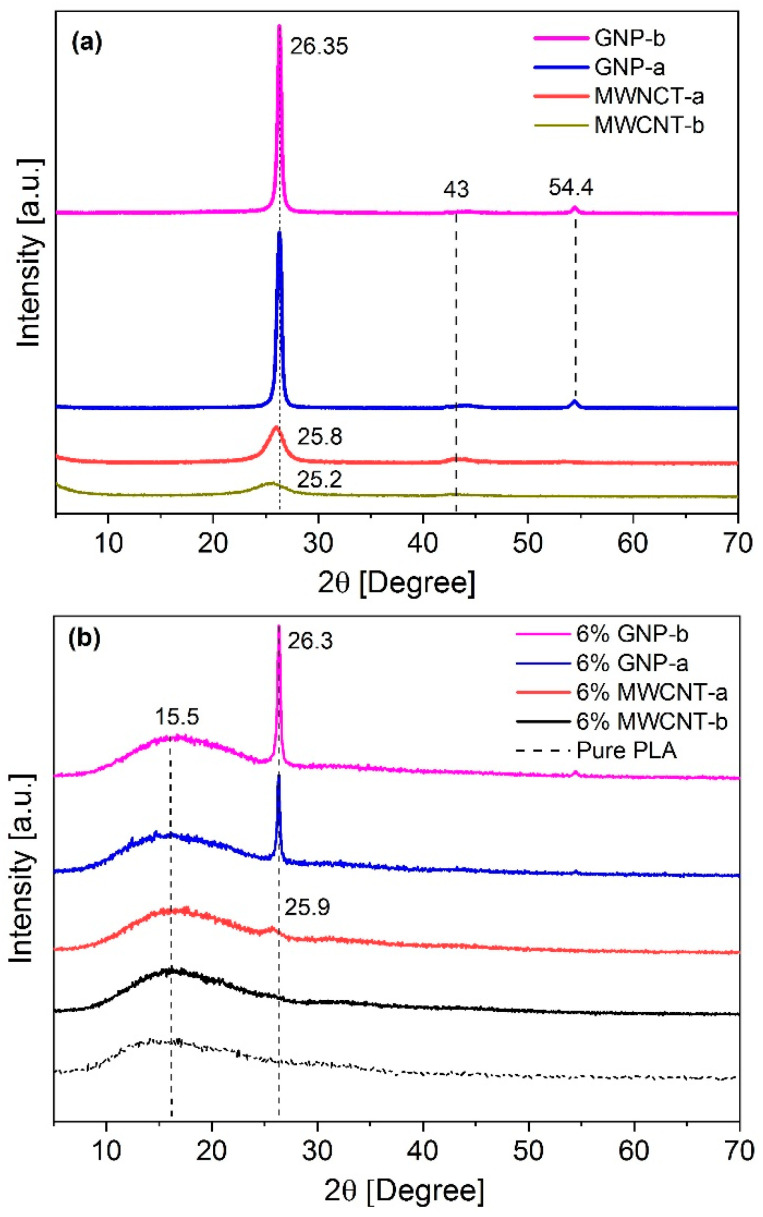
WAXD diffraction of (**a**) fillers, graphene (GNP-a and GNP-b) and carbon nanotubes. (MWCNT-a and MWCNT-b); (**b**) nanocomposites with 6 wt.% fillers of GNP-a, GNP-b, MWCNT-a, and MWCNT-b with the neat PLA, as a reference.

**Figure 5 polymers-12-01208-f005:**
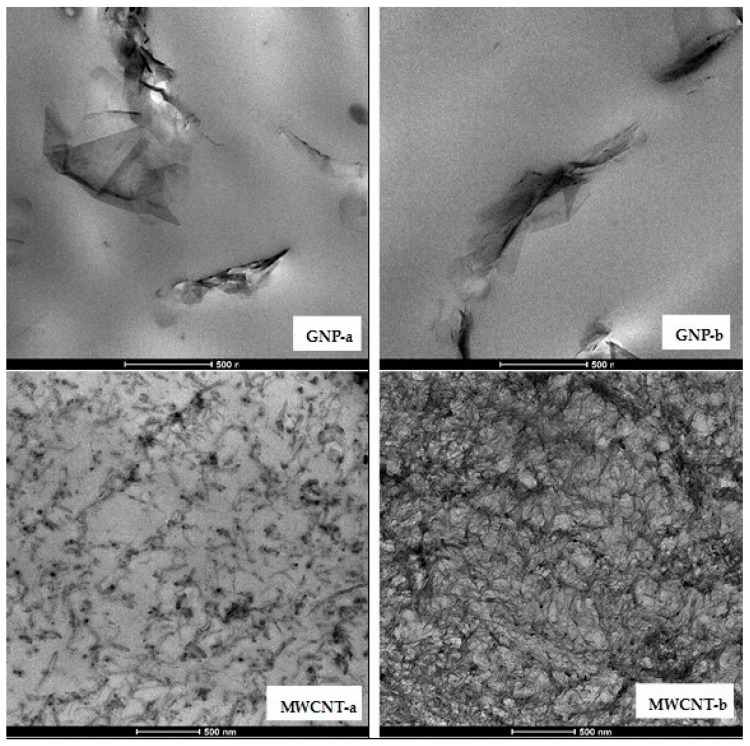
TEM micrographs of 6 wt.% nanocomposites, comparing GNP-a and GNP-b (first line); MWCNT-a and MWCNT-b (second line).

**Figure 6 polymers-12-01208-f006:**
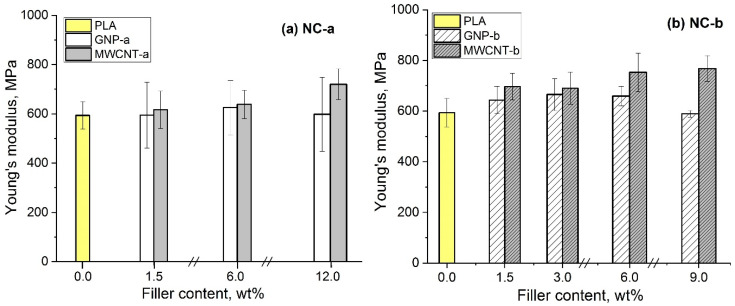
Young’s modulus vs. filler content of the neat PLA and nanocomposites with varying filler content. Comparison of nanocomposite series: (**a**) NC-a with GNP-a, MWCNT-a; and (**b**) NC-b with GNP-b, MWCNT-b.

**Figure 7 polymers-12-01208-f007:**
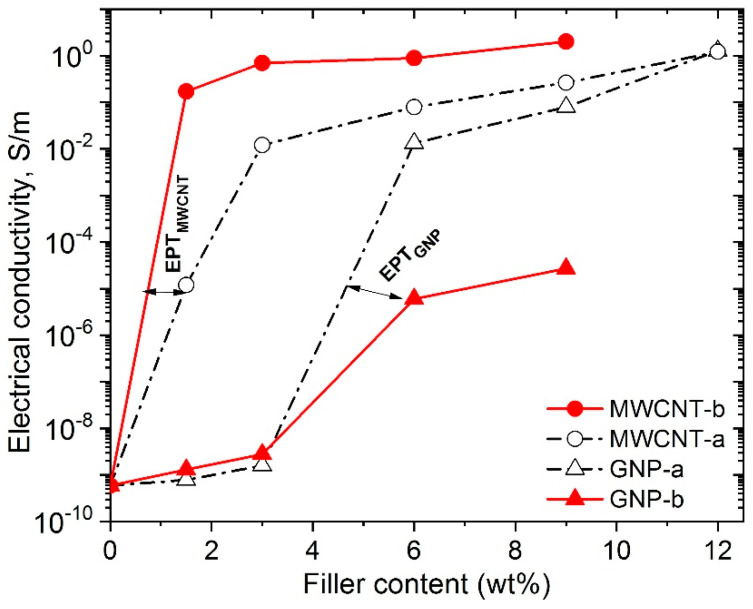
Electrical conductivity vs. filler content for the MWCNT-a, MWCNT-b, GNP-a and GNP-b nanocomposites. Arrows show EPT, which coincide with rheological percolation threshold.

**Figure 8 polymers-12-01208-f008:**
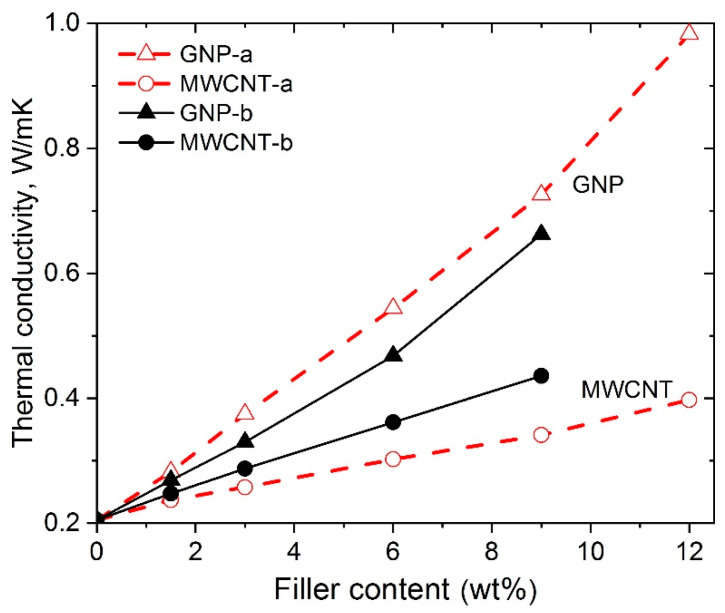
Thermal conductivity of nanocomposite series NC-a and NC-b, as varying the GNPs and MWCNTs contents.

**Figure 9 polymers-12-01208-f009:**
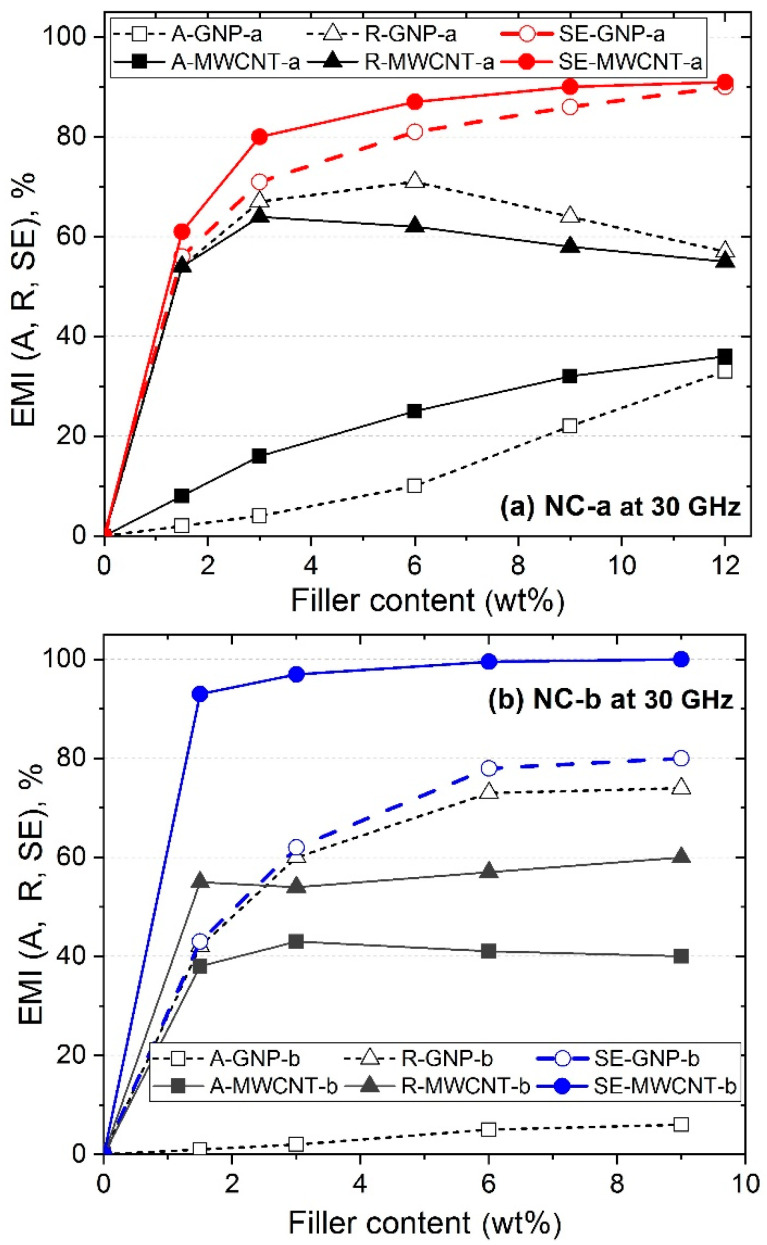
Reflection (R), absorption (A) and electromagnetic shielding efficiency (SE) of nanocomposites, with GNPs and MWCNTs, as varying the filler contents, for the PLA-based nanocomposite series: (**a**) NC-a and (**b**) NC-b.

**Table 1 polymers-12-01208-t001:** Characteristics of the two types of carbon nanofillers, GNPs and MWCNTs.

Characteristics	GNP-a	GNP-b	MWCNT-a	MWCNT-b
Trade Name	TNIGNP	TNGNP	TNIMH4	NC7000
Supplier	Times Nano	Times Nano	Times Nano	Nanocyl
Purity, wt.%	90	>99.5	95	90
Layers	<30	<20	-	-
Thickness, nm	4–30	4–20	-	-
Average size, μm	5–7	5–10	-	-
Outer diameter, nm	-	-	10–30	9.5
Inner diameter, nm	-	-	5–10	~5
Length, μm	-	-	10–30	1.5
Aspect ratio	240	500	1000	150
Functionalization (-OH, %)	-	-	2.48	oxidized
Specific Surface Area, SSA m^2^/g	30–40	30–40	>110	250–300
Volume Resistivity, Ohm.cm	<0.15	4.10^−4^	>10^−2^	10^−4^

**Table 2 polymers-12-01208-t002:** Viscosity at $\dot{\gamma }$ = 0.05 s^−1^, flow index (n) and other dispersion characteristics of NC-a and NC-b, filled with 1.5 and 6 wt.% GNP and MWCNT.

Sample	Viscosity at $\dot{\gamma }$ = 0.05 s^−1^ φ = 1.5 wt.% [MPa.s]	Flow Index (*n*) at φ = 1.5 wt.%	Viscosity $\dot{\gamma }$ = 0.05 s^−1^ φ = 6 wt.% [MPa.s]	Flow Index (*n*), at φ = 6 wt.%	Intrinsic viscosity [η] Equation (2)	RPT Equation (2) [wt.%]	EPT Equation (6) [wt.%]	*m* Equation (3)	*D*	*H*(*τ*) at *τ* = 50 s
PLA	0.81	0.991	0.81	0.991	-	-	-	-	-	1E-4
GNP-a	1.24	0.930	3.99	0.737	35	5	5	4.39	1.861	12
GNP-b	0.97	0.905	3.65	0.697	114	6	6	4.53	1.896	0.4
MWCNT-a	7.37	0.466	66.34	0.201	329	1.5	1.4	2.77	1.195	9E3
MWCNT-b	42.17	0.339	796.38	0	5001	0.5	0.5	2.96	1.311	1.2E5

**Table 3 polymers-12-01208-t003:** Tensile mechanical characteristics of NC-a and NC-b nanocomposites, as varying the filler contents.

Filament Samples NC-a	Young’s Modulus MPa	Ultimate Strength MPa	Elongation %	Filament Samples NC-b	Young’s Modulus MPa	Ultimate Strength MPa	Elongation %
PLA	594 ± 16	28.0 ± 3.9	8.5 ± 0.5	PLA	594 ± 16	28.0 ± 4.0	8.5 ± 0.5
1.5GNP-a	575 ± 34	23.0 ± 3.5	7.2 ± 11	1.5GNP-b	644 ± 54	27.5 ± 1.5	7.9 ± 1.0
6GNP-a	625 ± 110	20.9 ± 2.3	6.0 ± 0.4	3GNP-b	665 ± 63	26.6 ± 2.6	7.0 ± 0.6
12GNP-a	598 ± 159	19.9 ± 3.3	5.6 ± 0.7	6GNP-b	659 ± 39	23.9 ± 1.2	6.6 ± 1.3
1.5MWCNT-a	616 ± 76	25.8 ± 2.3	7.2 ± 0.5	9GNP-b	598 ± 13	24.8 ± 1.2	6.8 ± 1.7
3 MWCNT-a	599 ± 49	26.6 ± 0.9	9.0 ± 0.9	1.5MWCNT-b	697 ± 52	29.1 ± 0.8	8.2 ± 1.3
6 MWCNT-a	638 ± 58	27.9 ± 2.4	7.2 ± 0.5	3MWCNT-b	690 ± 64	31.7 ± 3.8	9.0 ± 1.3
9 MWCNT-a	674 ± 73	26.0 ± 1.9	5.4 ± 0.7	6MWCNT-b	756 ± 76	36.7 ± 4.3	9.2 ± 1.1
12 MWCNT-a	720 ± 63	23.3 ± 1.9	4.3 ± 0.4	9MWCNT-b	768 ± 51	27.2 ± 3.2	6.1 ± 1.1

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
