# Peer review of "Essential Nanostructure Parameters to Govern Reinforcement and Functionality of Poly(lactic) Acid Nanocomposites with Graphene and Carbon Nanotubes for 3D Printing Application"

_polymers, 2020, doi:10.3390/polym12061208_

Round 1

Reviewer 1 Report

The article (Polymers-806385) focuses on the topic of the effect of graphene and carbon nanotubes on the mechanical, electrical, thermal, and electromagnetic properties of poly(lactic) acid. This is an important topic, and which is suitable for the journal of POLYMERS. It is obviously that the authors have carried out lots of experimental works and theoretical analysis. It is worthy to be published. However, for the benefit of the readers, a moderate revision as listed below are still be suggested before its publishing.

1) One of the major concerns is the 3-D printing is not done at all in this article. After reading the article, it is obviously that the authors want to express the feasibility of the modified PLA as FDM materials. Although the modification seems to be successful by plenty of tests, it is not still clear whether the modification of PLA using the graphene and carbon nanotubes can improve the properties of 3-D printed samples, since all the samples of test and characterization in this article are not prepared by 3-D printing. Authors have published many articles about the 3-D printing application of PLA, as mentioned in reference [10-15]. If the authors do not want to supply the experiments, at least some results should be listed. As a result, in the conclusion, the word “may be” (L551) can be changed to be a more certain word.

2) More discussion about the 3-D printing are strongly suggested to be provided. It is actually closely related to the first concern. The title of this article includes the words of “for 3D printing application”, and it is also not suggested to be deleted. Although the article mainly focuses on the basic structural parameters of the modified materials, the 3-D printing should not yet be ignored. The 3-D printing process can influence and even decide the properties of 3-D printed parts. For example, printing induced effect was reported, Rapid Prototyping Journal, 2017, 23, 869-880, and Industrial & Engineering Chemistry Research, 2019, 58, 47, 21476-21484. Hence, this reviewer suggests the authors are better to mention it and address the concern adequately.

3) The figures in this article were carefully prepared. However, some minor revisions are suggested. In figure 2, please revise the blue line to make it clearer. In figure 4, the gray line is also suggested to be clearer. In figure 6, please move the bar to right location indicated by the filler content, and the distance between two bars are better to be same.

4) The results of elongation tested should be at lease two effective number, why it is 8±0.5? It is exactly 8.0? If so, 8.0±0.5 should be better.

5) The results of tensile strength should be also at least two effective number. For example, 28±4 revises to be 28.0±4.2. In addition, the strength of PLA seems to be somewhat low, and a comparison are better to be provided.

6) The contents in this article were carefully edited by authors. However, the used symbols should be united. For example, τ should be keeping in italic (P7L243). In addition, the symbol of full stop is missed (L244). Check it more carefully.

Author Response

The article (Polymers-806385) focuses on the topic of the effect of graphene and carbon nanotubes on the mechanical, electrical, thermal, and electromagnetic properties of poly(lactic) acid. This is an important topic, and which is suitable for the journal of POLYMERS. It is obviously that the authors have carried out lots of experimental works and theoretical analysis. It is worthy to be published. However, for the benefit of the readers, a moderate revision as listed below are still be suggested before its publishing.

A. Authors are thankful to the valuable comments of the Reviewer 1. Our answers are listed below, as well as the respective corrections in the text are marked in green background in the revised manuscript.

Q1) One of the major concerns is the 3-D printing is not done at all in this article. After reading the article, it is obviously that the authors want to express the feasibility of the modified PLA as FDM materials. Although the modification seems to be successful by plenty of tests, it is not still clear whether the modification of PLA using the graphene and carbon nanotubes can improve the properties of 3-D printed samples, since all the samples of test and characterization in this article are not prepared by 3-D printing. Authors have published many articles about the 3-D printing application of PLA, as mentioned in reference [10-15]. If the authors do not want to supply the experiments, at least some results should be listed. As a result, in the conclusion, the word “may be” (L551) can be changed to be a more certain word.

A1. Authors agree with the Reviewer comment that the 3D printing is not reported in this article, as the goal was to study the main factors influencing the design of the novel nanocomposites material suitable for 3D printing, but not the 3D printing process itself. However, in our previous studies [Ref. 10-15, 22,23 ] the 3D printing process, as well as and the characteristics of nanocomposite filament and the 3D printed samples were reported in more details. Therefore, answering the reviewer comments we propose the following changes in the text of the manuscript.

      -  In lines 69 – 73, a new text is added:

In our previous studies [10-15] we have reported on the development of innovative PLA-based nanocomposites incorporating graphene nanoplatelets and multiwall carbon nanotubes, suitable for 3D printing applications. The present study aims to identify the essential structural parameters for the design of enhanced mechanical, electrical, electromagnetic and thermal properties of nanocomposites, compared to the neat PLA.”

- In Lines 73-76, this sentence is deleted: This study aims to identify the essential structural parameters affecting different properties of the novel multifunctional poly(lactic) acid–based nanocomposites filled with graphene nanoplatelets and carbon nanotubes, suitable for 3D printing applications [10-15].

- In Line 586, “may be” is deleted and “are” is added

Q2) More discussion about the 3-D printing are strongly suggested to be provided. It is actually closely related to the first concern. The title of this article includes the words of “for 3D printing application”, and it is also not suggested to be deleted. Although the article mainly focuses on the basic structural parameters of the modified materials, the 3-D printing should not yet be ignored. The 3-D printing process can influence and even decide the properties of 3-D printed parts. For example, printing induced effect was reported, Rapid Prototyping Journal, 2017, 23, 869-880, and Industrial & Engineering Chemistry Research, 2019, 58, 47, 21476-21484. Hence, this reviewer suggests the authors are better to mention it and address the concern adequately.

A2. Following the reviewer’s suggestion, we have listed some results on the 3D printing process from our previous studies:

- In Lines 547 – 565, the following new text is added:

The applicability of PLA-based nanocomposites filled with GNPs and MWCNTs was verified to FDM technology in our previous studies [10-15, 22,23]. The printability window of the nanocomposite filament was identified to be similar to that of the neat PLA for the percolated nanocomposites, while at filler contents highly above the percolation threshold, the printability window is shifted towards higher printing rates. A comparison was made between the extruded filament, the 3D printed samples and the hot pressed one [22] in respect to their electrical conductivity and tensile properties. It was found that the filament demonstrate 2-3 decades higher electrical conductivity and 50-150% higher tensile mechanical characteristics, compared to that of 3D printed and hot-pressed samples, due obviously to some preferential orientation of anisotropic carbon nanoparticles during the filament extrusion, which favor the percolation network. Moreover, the 3D printed samples demonstrate almost 20–50% higher crystallinity compared to the hot-pressed samples, however, the enhanced crystallinity has insufficient effect on the tensile properties. Our results also indicate that there is a close relationship between mechanical properties and morphological structure of nanocomposites, both deeply influenced by the type, the concentration and the degree of dispersion of nanofiller. Recently, researchers [36, 37] reported that the deposition orientation has a significant influence on the mechanical behavior of the FDM 3D printed parts, compared to the filament. All these aspects may be associated with anisotropy of electrical and mechanical properties attributed to the nanofiller orientation in the matrix polymer and influenced by the deposition-induced effects during FDM. These effects of 3D printing will be investigated in more details in our future studies.”

  • In Lines 700-704, the new references are added [36] and [37].

Q3) The figures in this article were carefully prepared. However, some minor revisions are suggested. In figure 2, please revise the blue line to make it clearer. In figure 4, the gray line is also suggested to be clearer. In figure 6, please move the bar to right location indicated by the filler content, and the distance between two bars are better to be same.

A3. The required corrections are made in Figures 2, 4 and 6

Q4) The results of elongation tested should be at lease two effective number, why it is 8±0.5? It is exactly 8.0? If so, 8.0±0.5 should be better.

A4. The required corrections are made in Table 3

Q5) The results of tensile strength should be also at least two effective number. For example, 28±4 revises to be 28.0±4.2. In addition, the strength of PLA seems to be somewhat low, and a comparison are better to be provided.

A5. The required corrections are made in Table 3

Q6) The contents in this article were carefully edited by authors. However, the used symbols should be united. For example, τ should be keeping in italic (P7L243). In addition, the symbol of full stop is missed (L244). Check it more carefully.

A6. The required corrections are made in Lines 263 – 264.

Reviewer 2 Report

The paper by Rumiana Kotsilkov et al reports an interesting study on the effect of carbon-based fillers on the mechanical, electrical and thermal properties of PLA composites.

Overall, the paper is interesting, however, it should be improved in the parts described below:

  • Rheological parameters: it is well known that the mechanical stresses and the temperature can cause degradation of PLA, and this is even more true when carbon based fillers are adopted (see e.g. http://dx.doi.org/10.3390/polym10020139). The authors should comment on this point, because all the rheological behavior can be affected by degradation
  • a flow index close to 1 suggests a Bingham-like behavior. Is that expected for a filler content of 1.5%?
  • the analysis reported in figure 2a is unclear and should be better explained. Where do the continuous lines come from and in which way the dotted lines are obtained?
  • the wavy behaviour of  GNP-b in figure 3 needs to be commented (possibly the test should be repeated)

Author Response

Response to Reviewer 2

The paper by Rumiana Kotsilkova et al reports an interesting study on the effect of carbon-based fillers on the mechanical, electrical and thermal properties of PLA composites. Overall, the paper is interesting, however, it should be improved in the parts described below:

A.  Authors thank to the valuable comments of the Reviewer 2. Our answers are listed below, as well as the respective corrections in the text are marked in yellow background in the revised manuscript.

Q1. Rheological parameters: it is well known that the mechanical stresses and the temperature can cause degradation of PLA, and this is even more true when carbon based fillers are adopted (see e.g. http://dx.doi.org/10.3390/polym10020139). The authors should comment on this point, because all the rheological behavior can be affected by degradation.

A1. Authors agree with the reviewer comment that the mechanical stresses and the temperature during extrusion for long time may cause partial degradation of the biodegradable PLA polymer. As shown by D’Urso L., et all. (http://dx.doi.org/10.3390/polym10020139), the addition of carbon nanofillers (e.g. graphene) stabilize PLA towards degradation reaction. Similar effect was found in our study, Ref [23], where the graphene and carbon nanotube filers suppress the aging decomposition reactions of PLA, moreover the structure and mechanical properties of aged nanocompisutes may be improved by annealing and reprocessing. In the present manuscript, the PLA and the nanocomposites were prepared by two extrusion runs, taking few minutes of each, at low temperatures of 170–180 â—¦C. The short processing time and the low processing temperatures are expected to prevent the PLA degradation. Moreover, The addition of carbon-based fillers makes nanocomposites  more stable to degradation during such extrusion conditions, compared to the neat PLA. Based on this, we have ignored the effect of polymer degradation on rheological behavior in our study.

In lines 106-109 of the manuscript, the following text is added:

The short processing time, the low processing temperatures and the addition of carbon nanofillers prevent the PLA degradation. Therefore, the decomposition processes are insufficient during the extrusion of nanocomposite filament, thereby insufficient effect of polymer degradation on rheological results is expected.”

Q2. A flow index close to 1 suggests a Bingham-like behavior. Is that expected for a filler content of 1.5%?

A2. From the viscosity model  (Eq. 1) the power law slope is presented by (n-1), where (n) is the flow index.

In lines 173 - 177, a new text is added:

In the case of the neat PLA and the 1.5wt% GNP composites, the power law slope is of  (n-1)0, which determine the flow index of (n1), therefore a Newtonian-like behavior. While for the 1.5 wt% MWCNT composites, as well as for all composites with 6 wt% filler content, a Bingham-like behavior is observed, having power law slope of Eq. 1, (n - 1) >> 0, so the flow index is of (n) < 0.5, as shown in Table 2.”

Q3. The analysis reported in figure 2a is unclear and should be better explained. Where do the continuous lines come from and in which way the dotted lines are obtained?

A3: The following corrections are made:

  • In Line 195. The title of Table 2 title is changed to: “Table 2. Viscosity at =0.05 s-1 , flow index (n) and other dispersion characteristics of NC-a and NC-b, filled with 1.5 and 6 wt% GNP and MWCNT”

      -      In Table 2, a new column is added, summarizing the values of the intrinsic viscosity [η], calculated from the experimental data of the relative viscosity and the Eq. 2. The relaxation spectra weigh H(τ) is changed for the 6wt% GNP-b nanocomposites

-   In Lines 212 - 214, a new text is added, as follows: … “shown in Table 2. In Fig. 2(a) the continuous lines present the B-spline curve-fitting of the relative viscosity data at a fixed shear rate ), while the dotted lines show the fitting by the adapted Einstein type equation, Eq. (2)..”

Q4. The wavy behaviour of  GNP-b in figure 3 needs to be commented (possibly the test should be repeated).

A4. The test for the 6GNP-b composite was repeated and the relaxation spectra was recalculated in Fig. 3.

Round 2

Reviewer 1 Report

The authors have adequately and satisfactorily addressed my previous comments. I would like to suggest this article be accepted in the present form.

Reviewer 2 Report

The authors addressed all the points raised during the first review. The paper can now be accepted.